# Design of a Gene Panel to Expose the Versatile Role of Hepatic Stellate Cells in Human Liver Fibrosis

**DOI:** 10.3390/pharmaceutics12030278

**Published:** 2020-03-20

**Authors:** Fransien van Dijk, Christa M. Hazelhoff, Eduard Post, Gerian G. H. Prins, Krista Rombouts, Klaas Poelstra, Peter Olinga, Leonie Beljaars

**Affiliations:** 1Groningen Research Institute of Pharmacy, Department of Pharmacokinetics, Toxicology and Targeting, University of Groningen, 9713 AV Groningen, The Netherlands; fransien.van.dijk@rug.nl (F.v.D.); c.m.hazelhoff@student.rug.nl (C.M.H.); e.post@rug.nl (E.P.); k.poelstra@rug.nl (K.P.); 2Groningen Research Institute of Pharmacy, Department of Pharmaceutical Technology and Biopharmacy, University of Groningen, 9713 AV Groningen, The Netherlands; g.g.h.prins@rug.nl (G.G.H.P.); p.olinga@rug.nl (P.O.); 3UCL Institute for Liver and Digestive Health, University College London Division of Medicine, London NW3 2PF , UK; k.rombouts@ucl.ac.uk

**Keywords:** drug development, primary hepatic stellate cells, precision-cut liver slices, myofibroblast, reversion, fibrosis gene panel, transforming growth factor β, platelet-derived growth factor BB

## Abstract

The pivotal cell involved in the pathogenesis of liver fibrosis, i.e., the activated hepatic stellate cell (HSC), has a wide range of activities during the initiation, progression and even regression of the disease. These HSC-related activities encompass cellular activation, matrix synthesis and degradation, proliferation, contraction, chemotaxis and inflammatory signaling. When determining the in vitro and in vivo effectivity of novel antifibrotic therapies, the readout is currently mainly based on gene and protein levels of α-smooth muscle actin (α-SMA) and the fibrillar collagens (type I and III). We advocate for a more comprehensive approach in addition to these markers when screening potential antifibrotic drugs that interfere with HSCs. Therefore, we aimed to develop a gene panel for human in vitro and ex vivo drug screening models, addressing each of the HSC-activities with at least one gene, comprising, in total, 16 genes. We determined the gene expression in various human stellate cells, ranging from primary cells to cell lines with an HSC-origin, and human liver slices and stimulated them with two key profibrotic factors, i.e., transforming growth factor β (TGFβ) or platelet-derived growth factor BB (PDGF-BB). We demonstrated that freshly isolated HSCs showed the strongest and highest variety of responses to these profibrotic stimuli, in particular following PDGF-BB stimulation, while cell lines were limited in their responses. Moreover, we verified these gene expression profiles in human precision-cut liver slices and showed similarities with the TGFβ- and PDGF-BB-related fibrotic responses, as observed in the primary HSCs. With this study, we encourage researchers to get off the beaten track when testing antifibrotic compounds by including more HSC-related markers in their future work. This way, potential compounds will be screened more extensively, which might increase the likelihood of developing effective antifibrotic drugs.

## 1. Introduction

In liver fibrosis research, when screening the efficacy of a promising new antifibrotic compound, the current gold standards to assess the in vivo antifibrotic efficacy are α-smooth muscle actin (α-SMA) and collagens type I and III. α-SMA is highly expressed by the pathogenic-activated hepatic stellate cells (HSCs) that transform into myofibroblasts in the fibrotic liver [1]. In addition, the deposited interstitial collagen fibrils are predominantly produced by these cells [1]. Collagenases such as matrix metalloproteinases (MMPs) and certain profibrotic cytokines like transforming growth factor β (TGFβ) are sometimes added as markers for HSC activities [2]. However, the involvement of HSCs in fibrosis is very diverse, and thus only taking into account these few markers might leave other relevant characteristics of the HSCs underexposed.

Upon liver injury, HSCs are activated to develop into myofibroblasts following a response that is associated with the release of a variety of profibrogenic mediators, of which TGFβ and platelet-derived growth factor BB (PDGF-BB) are established as the central players [3]. Although other hepatic cells including portal fibroblasts, bone marrow-derived mesenchymal cells (fibrocytes) and epithelial/endothelial cells transition into mesenchymal cells (EMT/EndMT), and can contribute to the myofibroblast pool as well, their contribution in cell numbers is much lower [4,5,6,7]. It was recently assessed that 82–96% of the myofibroblast pool in models of toxic-, cholestatic- and fatty-liver disease originates from hepatic stellate cells [8]. Following the initiation of HSC activation and transdifferentiation into myofibroblasts, perpetuation of the response occurs, indicating an amplification and expansion of the activated state. This is associated with phenotypic changes, altered matrix synthesis and degradation, increased proliferation, chemotaxis and contraction [3,9]. As a consequence, during this transition, HSCs typically lose their cytoplasmic lipid droplets, which mainly contain vitamin A compounds. When the hepatic injury and the subsequent inflammatory response persist, the fibrogenesis and matrix deposition becomes problematic and may progress from fibrosis to irreversible stages of cirrhosis [3].

It is nowadays commonly accepted that fibrosis, even in advanced stages, is reversible upon eradication of the inciting stimulus or can be induced by (antifibrotic) drugs [10]. The exact mechanisms of reversion are not elucidated yet, but a crucial event involves the regression of activated HSCs and myofibroblasts [9]. This might occur via apoptosis, cellular senescence, clearance by immune cells or reversion to a less activated cell state [11]. These reverted or inactivated HSCs share various similarities with quiescent HSC. However, they also express different genes and are more primed to fibrogenic stimuli, such as TGFβ, as compared to quiescent cells [10,12,13].

Currently, patients suffering from end-stage liver disease can only be cured by organ transplantation, of which the availability is limited [14]. Therefore, effective antifibrotic drugs are urgently needed but the success of the clinical trials is low and needs long-term evaluations. To improve and accelerate drug development, reliable and comprehensive in vitro screening methods to test putative compounds are indispensable. Moreover, these screening methods should primarily focus on human models [14]. The preclinical outcomes in animal models do not always correlate well with the human situation due to species differences, and this is one of the causes of unnecessary preclinical rejection of potential compounds [15].

Since the activated HSCs are an important target for future drug intervention of hepatic fibrosis, in our current study we focused on this pivotal cell type with in vitro studies in primary human cells and commonly used human HSC lines, and with ex vivo studies using precision-cut liver slices (PCLS). PCLS, freshly prepared from residual human livers, represent an ex vivo tissue culture technique that retains the multicellular characteristics of the hepatic (fibrotic) environment in vivo [15]. In our studies, we assessed the HSC activities in response to the two key profibrotic factors, TGFβ and PDGF-BB. While TGFβ is predominantly known for its role in cellular activation and transformation, and in matrix biology, PDGF-BB is more involved in the survival, proliferation and chemotaxis of HSCs [16]. In the used in vitro and ex vivo human models, we looked beyond α-SMA and collagens, typically regulated by TGFβ, and we now present a panel of genes related to the versatile HSC activities in fibrogenesis and fibrolysis, thereby providing a framework that could improve antifibrotic drug development.

## 2. Materials and Methods

### 2.1. Cell Cultures

Human livers were obtained under the University College London—Royal Free BioBank Ethical Review Committee, reference number: NC2015.020 (B-ERC-RF) after informed consent was obtained for each donor. Primary human hepatic stellate cells (pHSC) were freshly isolated from these human livers as described before [17] and cultured in Iscove’s Modified Dulbecco’s Medium, supplemented with 20% fetal bovine serum (FBS) performance plus, 100 U/mL penicillin, 100 μg/mL streptomycin, 2 mM l-glutamine, 1 mM sodium pyruvate, 1× Minimal Essential Medium (MEM) non-essential amino acids solution and 250 ng/mL amphotericin B. The Human Hepatic Stellate Cells (HHSteC) were obtained from ScienCell Research Laboratories (Carlsbad, CA, USA), and cultured in poly-l-lysine (Sigma-Aldrich, St. Louis, MO, USA)-coated flasks and plates in complete stellate cell medium containing 2% FBS, 100 U/mL penicillin, 100 μg/mL streptomycin and 1% stellate cell growth supplement (all ScienCell Research Laboratories). HHSteCs and pHSCs were sub-cultured up to 5 times, ensuring minimal variation in the myofibroblast activation state. LX-2 hepatic stellate cells were kindly provided by prof. Scott Friedman (Mount Sinai Hospital, New York, NY, USA) [18]. Cells were cultured in Dulbecco’s modified Eagle’s medium (DMEM, GlutaMAX™) containing 10% FBS, supplemented with 100 U/mL penicillin and 100 μg/mL streptomycin. TWNT-4 hepatic stellate cells were cultured in DMEM containing 10% FBS, 100 U/mL penicillin, 100 μg/mL streptomycin and 2 mM·L-glutamine [19].

Cells were plated in 12-well plates (for pHSC 70,000 cells/well, for HHSteC 125,000 cells/well, and for LX-2 and TWNT-4 50,000 cells/well) and either not stimulated (control), or stimulated for 24 h with 5 ng/mL TGFβ1 or 50 ng/mL PDGF-BB (both PeproTech, London, Great Britain), and harvested for RNA isolation and cDNA synthesis. Materials were purchased at Thermo Fisher Scientific (Waltham, MA, USA) unless stated otherwise. Photomicrographs of the different cell types were captured at 40× and 100× magnification.

### 2.2. Precision-Cut Liver Slices (PCLS)

Healthy human liver tissue was obtained from donor organs unsuitable for transplantation, resized donor organs and unaffected tissue of patients undergoing surgical excision of carcinoma. The experimental protocols were approved by the Medical Ethical Committee of the University Medical Center Groningen. Precision-cut liver slices were prepared according to standard procedures, as described previously [15,20]. In short, livers (*n* = 4) were sliced in enhanced ice-cold carbogen-saturated Krebs–Henseleit buffer and cut with a Krumdieck tissue slicer (Alabama R&D, Munford, AL, USA) at 250–300 μm thickness and 4–5 mg wet weight. Slices were stored in University of Wisconsin (UW) until incubation for 48 h in 12-well plates filled with 1.3 mL William’s E Medium (Thermo Fisher Scientific) enriched with 25 mM glucose (Merck, Kenilworth, NJ, US) and 50 μg/mL gentamycin (Thermo Fisher Scientific), either not supplemented (control) or supplemented with 5 ng/mL TGFβ1 or 50 ng/mL PDGF-BB (both PeproTech). Culture medium was refreshed every 24 h. Wells of slices incubated with TGFβ1 were precoated with 10% bovine serum albumin (BSA) solution (Sigma-Aldrich). Per condition, 3 slices were pooled and snap-frozen for RNA isolation and cDNA synthesis.

### 2.3. Quantitative Real-Time Polymerase Chain Reaction (PCR)

Total RNA was isolated from cells or precision-cut liver slices using a Maxwell^®^ LEV simply RNA Cells/Tissue kit (Promega, Madison, WI, USA) according to the manufacturer’s instructions. RNA concentrations were determined using a NanoDrop One spectrophotometer (Thermo Fisher Scientific). Conversion of RNA to cDNA was performed using MLV (murine leukemia virus) reverse transcriptase (Promega) in an Eppendorf Mastercyler gradient device, with the gradient at 20 °C for 10 min, 42 °C for 30 min, 20 °C for 12 min, 99 °C for 5 min and 20 °C for 5 min. The transcription levels were measured in 10 ng cDNA by quantitative real-time PCR (SensiMix™ SYBR^®^ Low-ROX Kit, Bioline, Taunton, MA, USA) using a QuantStudio 7 Flex Real-Time PCR system (hold stage: 95 °C for 10 min; PCR stage: 40 cycles of 95 °C for 15 s and 60 °C for 25 s; melt curve stage: 95 °C for 15 s, 60 °C for 1 min and 95 °C for 15 s). Data was analyzed using QuantStudio Real-Time PCR software (Thermo Fisher Scientific). For each model, mRNA expression was normalized to housekeeping genes (either *ACTB* for primary HSC and LX-2, or *RNA18S5* for HHSteC, TWNT-4 and PCLS), and expressed as 2^−ΔΔCt^ (fold induction). Differences between treatment groups and untreated controls are in the main text referred to as percentage increase or decrease. The used primer sequences are listed in Table 1.

### 2.4. Statistical Analyses

At least 3 individual experiments were done, and these data are represented as mean ± standard error of the mean (SEM). The graphs and statistical analyses were performed in GraphPad Prism version 8.0 (GraphPad Software, San Diego, CA, USA). Statistical differences were assessed on 2^−ΔΔCt^ values by the Kruskal–Wallis test followed by Dunn’s multiple comparison test.

## 3. Results

### 3.1. Morphology of the in Vitro Models

We evaluated the gene panel in human HSCs with diverse backgrounds. In addition to primary HSCs, both freshly isolated from human livers (primary HSC) and obtained from a commercial source (HHSteC), we also include the immortalized HSC-lines LX-2 and TWNT-4 in our studies. First, these HSCs were examined for their morphology. All cultures displayed more or less the characteristic stellate star shape (Figure 1). The freshly isolated primary HSCs were bigger and had more elongated cell bodies compared to the other cell types. Upon culturing, all cells proliferated and became fully activated, which is characterized by their myofibroblast-like stretched polygon morphologies, and as a result, they lost their lipid droplet content.

### 3.2. TGFβ- and PDGF-BB-Related Fibrotic Responses in Primary Human Hepatic Stellate Cells

All HSCs were stimulated with either TGFβ or PDGF-BB, after which we assessed the expression of a panel of HSC-associated genes during fibrosis. The results classified according to the type of activity are shown in Figure 2. In the primary HSCs, several activation and fibrogenesis genes were significantly increased following stimulation with TGFβ (*ACTA2* 71.8% ± 12.8%, *CCN2* 125.2% ± 12.8%, *COL1A1* 46.4% ± 7.7%, *FN1* 49.2% ± 15.5%), as well as in HHSteCs (*ACTA2* 410.8% ± 78.9%, *CCN2* 702.5% ± 299.4%, *COL1A1* 170.5% ± 65.2%, *FN1* 264.4% ± 81.7%) (Figure 2A,B). In the HHSteCs, the expression of *TIMP1* was also significantly increased (30.6% ± 8.5%) (Figure 2C). Interestingly, expression of the contractility markers *MYLK* (62.7% ± 2.6%) and *ROCK2* (34.2% ± 3.9%) in primary HSCs (Figure 2E) was markedly reduced. Similarly, the reversion was assessed with the HSC quiescent markers *CYGB* and *PPARG*. The expression of *CYGB* (39.5% ± 2.4% and 77.6% ± 7.2% in primary HSCs and HHSteCs, respectively) and *PPARG* (68.1% ± 12.9% in HHSteCs) was significantly lower compared to cells not stimulated with TGFβ (Figure 2H), indicating the activation of the HSCs.

Upon stimulation with PDGF-BB, proliferation genes were markedly increased in both the primary HSCs (*CCND1* 78.6% ± 15.1%, *MKI67* 151.2% ± 49.8%) and the HHSteCs (*CCND1* 105.9% ± 32.7%, *MKI67* 79.4% ± 25.4%) (Figure 2D). The inflammatory gene *CXCL8* was increased as well (598.9% ± 247.0% in primary HSCs and 1743.4% ± 684.1% in HHSteCs, respectively) (Figure 2G). Moreover, *TIMP1* and *CCL2* expression were clearly induced by PDGF-BB in the primary HSC cultures (40.6% ± 12.6% and 79.8% ± 20.7%) (Figure 2C,F). Remarkably, several activation genes that were increased after TGFβ stimulation were significantly expressed lower in primary HSCs (*ACTA2* 64.9% ± 3.85%, *PDGFRB* 45.8% ± 2.7%, *CCN2* 26.8% ± 4.8%) and in HHSteCs (*PDGFRB* 43.9% ± 9.31%) after incubation with PDGF-BB (Figure 2A). More genes were responsive in primary HSCs as compared to HHSteCs, particularly following PDGF-BB stimulation, showing the added value of these primary cells.

Both primary HSC types showed stronger responses to either TGFβ- or PDGF-BB-stimulation as compared to the HSC-lines LX-2 and TWNT-4. LX-2 cells did show a clear increase in activation and fibrogenesis genes upon TGFβ-stimulation (*ACTA2* 62.4% ± 12.3%, *PDGFRB* 86.0% ± 10.7%, *CCN2* 150.6% ± 18.0%, *COL1A1* 165.3% ± 6.2%, *FN1* 139.2% ± 15.1%), whereas the gene expression of *COL1A1* was increased in TWNT-4 cells (91.3% ± 28.7%) (Appendix A). Of note, both cell lines showed hardly any effect on gene expression after PDGF-BB stimulation (Appendix A), and therefore the suitability of these cell lines for fibrosis research should be carefully considered. We summarized the differences in gene expressions of all cells in a heatmap, in which increased expressions as compared to control are depicted in red, and reduced expressions are shown in green (Table 2). Cycle threshold (Ct) values of the untreated controls of each gene in all in vitro models used are shown in Appendix A in order to demonstrate the differences in basal gene expression levels.

### 3.3. TGFβ- and PDGF-BB-Related Fibrotic Responses in Human Precision-Cut Liver Slices

To determine the gene expression levels of the different markers in a more comprehensive in vitro model, in which the in vivo liver environment is resembled, we also stimulated human precision-cut liver slices with either TGFβ or PDGF-BB. Although the change in expression was less pronounced as compared to the cell cultures, similar trends were found in the gene expressions (Figure 3). Stimulation of PCLS with TGFβ markedly increased the expression of *TIMP1* (137.4% ± 60.8%) and showed a clear increasing trend in expression of activation and fibrogenesis markers (*ACTA2* 189.6% ± 152.2%, *PDGFRB* 52.7% ± 53.2% and *COL1A1* 93.6% ± 69.9%) (Figure 3A–C). Interestingly, the expression of *PPARG* seemed to be reduced (23.1% ± 14.9%), which was again comparable to the in vitro results (Figure 3H).

Well in line with the in vitro results, PDGF-BB stimulation clearly increased the expression of proliferation markers *CCND1* and *MKI67* (86.4% ± 37.7% and 165.0% ± 47.3%), *TIMP1* (335.6% ± 118.0%) and *COL1A1* (180.6% ± 73.4%) (Figure 3B–D). Additionally, the chemotactic gene *CCL2* is expressed higher compared to the untreated control (99.9% ± 49.7%) (Figure 3F).

## 4. Discussion

The increasing incidence of patients with liver cirrhosis, combined with the lack of effective antifibrotic drugs to stop or reverse the disease, urgently demands drug development [14,21]. Clearly, activated HSCs have an important and versatile role in the development and progression of fibrosis of various etiologies, and are therefore an interesting therapeutic target [3,22]. Nowadays, several clinical trials focus on the HSC and study the effects of drugs that directly interfere with this cell type. Examples include the PPARy-agonist pioglitazone, and the dual CCR2/CCR5 receptor-antagonist cenicriviroc [3]. Despite the efforts, there is no curative treatment option for hepatic fibrosis on the market yet. When it comes to antifibrotic drug discovery and development, only evaluating the effect on α-smooth muscle actin (α-SMA) and collagens may give limited information. In fact, in our view, it is preferred to test promising antifibrotic compounds by a more extensive screening in human models with a variety of read-out parameters in order to boost drug discovery. We therefore developed a human gene panel that includes most of the previously defined HSC-activities during fibrosis. The panel accommodates, in total, 16 fibrotic HSC-related genes, which we tested on several human HSC-cell lines and primary cells, and on human precision-cut liver slices (PCLS) stimulated with one of the two main profibrotic mediators, i.e., transforming growth factor β (TGFβ) and platelet-derived growth factor BB (PDGF-BB). In the present study, we showed that freshly isolated HSCs have the strongest and most versatile response towards both profibrotic cytokines and is the most suitable system to test antifibrotic responses in future studies. In addition, liver slices, resembling the patient situation even better, showed analogous responses.

Following the activation of HSCs, the perpetuation phase is induced as the injury persists, which involves the proliferation, contraction, fibrogenesis, altered matrix degradation, chemotaxis and inflammatory signaling of the activated phenotype [3]. All these different aspects are typically regulated by either TGFβ or PDGF-BB and this can also be seen in the used primary HSCs (Figure 2) and slice model system (Figure 3). A multitude of other fibrogenic substances, such as connective tissue growth factor (CTGF), osteopontin (OPN), reactive oxygen species (ROS), tumor necrosis factor α (TNFα) and interleukin 1 β (IL-1β), contribute to the activation and proliferation of HSCs, but these are not as potent as TGFβ and PDGF-BB [23]. TGFβ characteristically promotes cellular activation (α-SMA) and increases extracellular matrix (ECM) deposition (including collagen type I and III and fibronectin), which is induced via intracellular Smad2/3 signaling [24]. TGFβ is also involved in the inhibition of matrix degradation, via suppression of matrix metalloproteinases (MMP1, 8, 13) and induction of tissue inhibitors of metalloproteinases (TIMP1, 2) and plasminogen activator inhibitor (PAI) [25]. PDGF-BB is the most potent growth factor essential in survival, proliferation and chemotaxis of HSCs via activation of Protein kinase B (PKB), also known as Akt, and extracellular signal-regulated kinase (ERK) 1/2 signaling [26]. In our studies, we clearly showed PDGF-induced expression of *CCND1, MKI67* and *CCL2* in primary HSCs and HHSteCs. Of note, in the HSC-lines that we studied as well (Appendix A), we could not detect changes in gene expression profiles after PDGF stimulation, in contrast to their preserved TGFβ responsiveness. This may show the limitations of these cell lines. Furthermore, PCLS responded well to PDGF with significantly increased expression of the corresponding parameters (Figure 3D–F) delineating the suitability of this ex vivo test system.

Removal of the fibrotic stimulus or treatment with an effective antifibrotic therapy would result in amelioration of hepatic fibrosis, associated with the resolution of the inflammatory response and the reduction and reversion of activated HSCs [27,28]. Studies in mice indicated that during resolution of liver fibrosis, approximately 50% of the activated HSCs escape from apoptosis and acquire an inactivated phenotype that is similar but not completely identical to quiescent HSCs [13]. These inactivated HSCs are in a primed state for reactivation in response to additional fibrogenic injury, and reacquire some, but not all characteristics of quiescent HSCs present in the healthy liver [12,13]. Here, we proposed *PPARG* and *CYGB* as markers for the quiescence of HSCs [29,30], and showed in the primary HSCs that these gene expressions are reduced predominantly by TGFβ stimulation, i.e., upon TGFβ-induced HSC activation. It is to be expected that the expression levels of these genes are increased when drugs with antifibrotic potential are tested, indicating the reversion of the disease.

As previously mentioned, there is no effective therapy for fibrosis available on the market yet, despite several compounds that were shown to effectively reduce fibrotic parameters both in vitro and in animal models [31]. An important problem when in vitro and in vivo data from animal experiments are translated to the human situation is the interspecies differences [20]. Depending on the studied species, the fibrotic process including interactions between cells, cytokines, extracellular matrix proteins, as well as the intracellular signaling, can be different. Therefore, it is of major importance to test potential antifibrotic compounds in validated human systems.

Using monocultures of hepatic stellate cells of human origin, the role of the key pathogenic cell type involved in liver fibrogenesis and the effectivity of drugs can be studied in detail. In addition, this can be studied in PCLS as well, in which the HSCs are present in their original microenvironment [15]. We observed similar fibrotic responses in the PCLS as detected in the primary HSCs, albeit less pronounced. This is mainly attributed to the presence of several different cell types, all contributing to the total gene expression. Of note, other hepatic cell populations including hepatocytes, Kupffer cells and cholangiocytes might also respond to the profibrotic stimuli TGFβ and PDGF-BB, thereby biasing gene expression results. In particular, in PCLS, it is essential to select markers that are as exclusively expressed by the HSC-population as possible. Obviously, in combination with the naturally occurring variation in gene expression in human donor tissues, this means that higher repetition of experiments is necessary in order to obtain statistically significant differences. A major advantage of PCLS is that cell–cell interactions, for instance between HSCs and liver-resident macrophages (Kupffer cells), can be included in the studies as well.

In this study, we used genetic markers to screen for the distinct functions of HSCs in hepatic fibrosis, as selected from the large database of fibrosis markers available in literature. Evidently, the choice for the markers per HSC function is open for interpretation and should be adapted at one’s own discretion, depending on the mechanism of action of the potential antifibrotic compound. We intended to create a framework for preclinical antifibrotic drug testing at the gene level. In addition to the genetic markers, the complementary protein levels are interesting as well. Currently, a wide variety of markers to be used in future clinical trials is under investigation. Methods include, for example, immunohistochemical staining for collagens, α-SMA or PDGFβ-receptor to study liver histology, elastography to determine liver stiffness and serum fibrosis markers [14].

In conclusion, model test systems to screen the antifibrotic potential of novel therapeutics should include all aspects of HSC activities in fibrosis, and thus look beyond the standard markers such as α-SMA and fibrillary collagens. The used models should be human-based, like primary HSCs or precision-cut liver slices freshly isolated or prepared from human livers, as these models will better predict therapeutic effects in patients. This way, the outcomes of in vitro studies with antifibrotic compounds can be broadened, thereby boosting drug development in the fibrosis field.

## Figures and Tables

**Figure 1 pharmaceutics-12-00278-f001:**
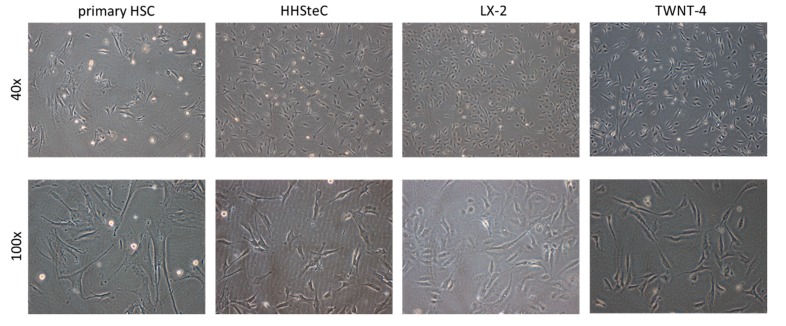
Morphology of various human stellate cell cultures (primary hepatic stellate cell (HSC), HHSteC, LX-2 and TWNT-4 cells). Representative light microscopy images were captured at 40× and 100× magnification at 1 day after seeding.

**Figure 2 pharmaceutics-12-00278-f002:**
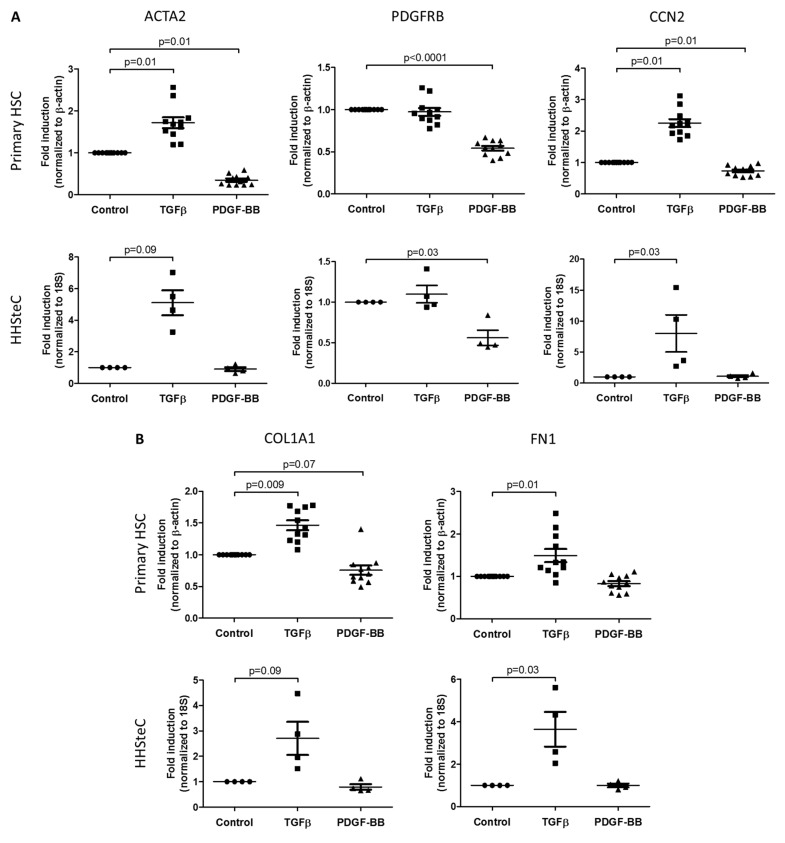
In vitro effects of TGFβ or PDGF-BB on the gene expression of markers for HSC-activities during fibrosis in primary HSCs (*n* = 11, 3 different donors) and HHSteCs (*n* = 4). HSC-activities and their associated genes include (**A**) activation (*ACTA2, PDGFRB, CCN2*), (**B**) fibrogenesis (*COL1A1, FN1*), (**C**) altered matrix degradation (*TIMP1*), (**D**) proliferation (*CCND1, MKI67*), (**E**) contractility (*MYLK, ROCK2*), (**F**) chemotaxis (*CCL2, CCL5*), (**G**) inflammatory signaling (*CXCL8*) and (**H**) reversion (*PPARG, CYGB*). The Ct values of *CCL5* and *PPARG* in primary HSCs and *TNF* in both cell types were not detectable. Fold inductions are relative to untreated controls.

**Figure 3 pharmaceutics-12-00278-f003:**
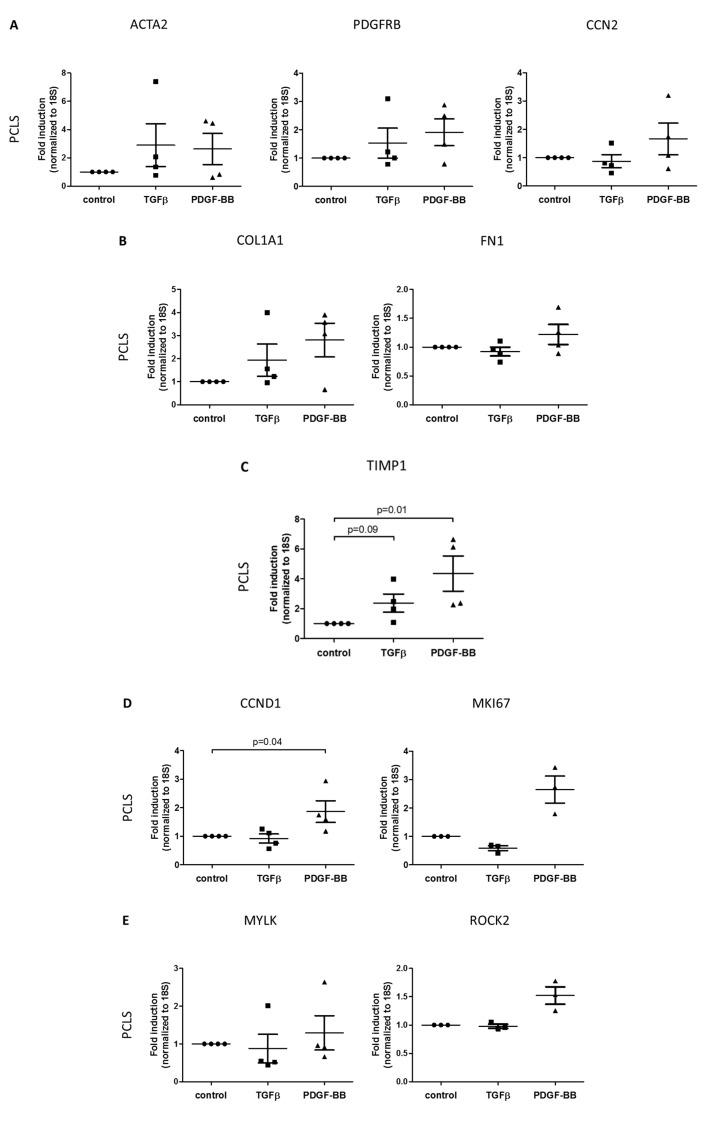
Gene expression levels of several markers for HSC-activities during fibrosis in human precision-cut liver slices (*n* = 4) stimulated with TGFβ or PDGF-BB. HSC activities and their associated genes include (**A**) activation (*ACTA2, PDGFRB, CCN2*), (**B**) fibrogenesis (*COL1A1, FN1*), (**C**) altered matrix degradation (*TIMP1*), (**D**) proliferation (*CCND1, MKI67*), (**E**) contractility (*MYLK, ROCK2*), (**F**) chemotaxis (*CCL2, CCL5*), (**G**) inflammatory signaling (*TNF, CXCL8*) and (**H**) reversion (*PPARG, CYGB*). Fold inductions are relative to untreated controls.

**Table 1 pharmaceutics-12-00278-t001:** Primer sequences used for quantitative real-time PCR.

Gene Symbol	Full Gene Name	Sequence Forward	Sequence Reverse
*ACTA2*	Alpha smooth muscle actin	CCCCATCTATGAGGGCTATG	CAGTGGCCATCTCATTTTCA
*PDGFRB*	Platelet-derived growth factor receptor beta	GTAAGGTGCCAACCTGCAAT	CATGGGGGTATGGTTTTGTC
*CCN2*	Cellular communication network factor 2	GACTGGAAGACACGTTTGGC	CTTCCAGGTCAGCTTCGCAA
*COL1A1*	Collagen type I alpha 1 chain	GTACTGGATTGACCCCAACC	CGCCATACTCGAACTGGAAT
*FN1*	Fibronectin 1	AGGCTTGAACCAACCTACGGATGA	GCCTAAGCACTGGCACAACAGTTT
*TIMP1*	Tissue inhibitor of metalloproteinases 1	AATTCCGACCTCGTCATCAG	TGCAGTTTTCCAGCAATGAG
*CCND1*	Cyclin D1	GACCCCGCACGATTTCATTG	AAGTTGTTGGGGCTCCTCAG
*MKI67*	Marker of proliferation Ki-67	CGTCCCAGTGGAAGAGTTGT	CCCCTTCCAAACAAGCAGGT
*MYLK*	Myosin light-chain kinase	GGTGACATGGCACAGAAACG	AGCTGCTTCGCAAAACTTCC
*ROCK2*	Rho-associated coiled coil-containing protein kinase 2	AACGTCAGGATGCAGATGGG	CAGCCAAAGAGTCCCGTTCA
*CCL2*	C-C motif chemokine ligand 2	CTCGCTCAGCCAGATGCAAT	TCCATGGAATCCTGAACCCAC
*CCL5*	C-C motif chemokine ligand 5	TGCTGCTTTGCCTACATTGC	CACACTTGGCGGTTCTTTCG
*TNF*	Tumor necrosis factor alpha	CGTCTCCTACCAGACCAAGG	CCAAAGTAGACCTGCCCAGA
*CXCL8*	C-X-C motif chemokine ligand 8	TGCAGTTTTGCCAAGGAGTG	CAACCCTCTGCACCCAGTTT
*PPARG*	Peroxisome proliferator activated receptor gamma	AGGAAGGGGCCTTAACCTCT	CACGGAGCTGATCCCAAAGT
*CYGB*	Cytoglobin	GTCATTCTGGAGGTGGTCGC	GTGGAGTTAGGGGTCCTACG
*ACTB*	Beta actin	CCTCGCCTTTGCCGATCC	AGGAATCCTTCTGACCCATGC
*RNA18S5*	RNA, 18S ribosomal N5	CGGCTACCCACATCCAAGGA	CCAATTACAGGGCCTCGAAA

**Table 2 pharmaceutics-12-00278-t002:** Heatmap visualizing the fold inductions of all cells studied and stimulated with TGFβ and PDGF-BB relative to the untreated control. Rows represent genes related to the versatile HSC activities in fibrogenesis and fibrolysis. Columns represent the tested cell types. Red color indicates increased gene expression (dark red *p* < 0.05; light red 0.05 ≤ *p* < 0.10) and green indicates reduced expression (dark green *p* < 0.05; light green 0.05 ≤ *p* < 0.10) as compared to control. No significant effect or no gene expression are depicted in light or dark grey, respectively. Of note, TIMP1 has an inhibitory function on this HSC-activity.

HSC Activity	Gene	TGF-β	PDGF-BB
pHSC	HHSteC	LX-2	TWNT-4	pHSC	HHSteC	LX-2	TWNT-4
Activation	*ACTA2*								
	*PDGFRB*								
	*CCN2*								
Fibrogenesis	*COL1A1*								
	*FN1*								
Matrix degradation	*TIMP1*								
Proliferation	*CCND1*								
	*MKI67*								
Contractility	*MYLK*								
	*ROCK2*								
Chemotaxis	*CCL2*								
	*CCL5*								
Inflammatory signaling	*TNF*								
	*CXCL8*								
Reversion	*PPARG*								
	*CYGB*

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
