# Peer review of "Design of a Gene Panel to Expose the Versatile Role of Hepatic Stellate Cells in Human Liver Fibrosis"

_pharmaceutics, 2020, doi:10.3390/pharmaceutics12030278_

Round 1

Reviewer 1 Report

The authors aimed to design a new panel of markers for HSC-features that allow to expand the read out regarding the role of HSC in liver fibrosis and regeneration.  Therefore, primary human HSC from two different sources and two immortalized HSC-cell lines as well as human precision-cut liver slices  were stimulated with TGFβ- and PDGF-BB. A couple of genes were analyzed by RT-PCR in order to define activation, fibrogenesis, protease-inhibitory potential, proliferation, contractility, chemotaxis, inflammatory signaling, and reversion.

The study is well designed, experiments were designed straight forward, the manuscript is well written, and the results, which are presented largely appropriately, might allow to draw the conclusions which are presented. Nevertheless, the expansion of the analyzed panel and a more critical discussion as well as other modifications might improve the manuscript:

  1. In line 104 the authors state ´HHSteCs and pHSCs were subcultured up to 5 times.´. As far as I know, HSC differentiate into myofibroblastes during 10d on plastic surface. That would mean that different activation states of primary HSC were analyzed. Is this correct and do the authors think this might have caused some bias ? Regarding Figure 1: how did the primary cells look like after 4-times subculturing ?
  2. I suggest to include GFAP into the panel. GFAP is an established specific marker for resting HSC and early activated HSC, which could provide information about the proliferation of HSC (Liver Transpl 14:806-814, 2008).
  3. Furthermore, desmin is a widely used marker for HSC-acivation, which should be included into the panel of markers. The assessment of GFAP, desmin, and aSMA together might complement the analysis of HSC-activation.
  4. The authors state that the suggested panel of markers should be adapted at one’s own discretion. Another table highlighting the most prominent markers and citations for the stated functions of HSC might complement this statement and the information provided. Due to this, the selection of the presented markers might be discussed more objectively.
  5. TGFβ- and PDGF-BB could also stimulate the gene expression of at least some of the analyzed markers in other hepatic cell populations, e.g. hepatocytes, cholangiocytes, inflammatory cells, endothelial cells, etc.. The authors should define address these points and discuss the possible bias in the PCLS-experiments extensively. Moreover, additional experiments might strengthen the authors conclusions, e.g. by co-in-situ hybridization or co-immunostaining that could demonstrate that the analyzed factors were mainly activated in HSC and/or HSC-derived myofibroblasts.
  6. In addition to the number of cells that might contribute to the myofibroblast pool, also the relevant functional contribution of other cell types, e.g. fibrocytes, to liver fibrosis should be stated (Cells 2019, 8, 1210).
  7. Indication of cell type is missing in Fig. 2c and h.
  8. In Fig. 3 TIMP-1-inhibitory function might be indicated. The correct name of TIMP-1 and other genes should be stated in table 1 under ´Full gene name´.
  9. The arrangement of figures 2 and 4 or the presentation of these results might be stratified for a better overview.

Reviewer 2 Report

General comments

In the present study, Dr. van Dijk and colleagues explored a panel of genes for the study of the response of human hepatic cell lines upon fibrotic stimuli. The authors include different experimental set-ups including primary cells, cell lines, and PCLS, this wide experimental approach is really appreciated.

Fibrogenesis is a complex process and limiting the research to just a few markers (alpha-SMA and Collagens) might be misleading and too simplistic. Thus the data obtained in this study, including other genes involved in the process of HSC transformation, will be very helpful for the study of liver fibrosis and for testing new therapeutic agents.

Minor comments

Materials and Methods

Cell culture section: How did the authors choose the TGF-b and PDGF-BB experimental concentration and time? How did they exclude a different sensitivity to the stimuli among the different cell types?

Just a curiosity: LX2 cell line has been reported (PMID: 15591520 - Xu, L Gut 2005) that must be cultured in low FBS medium (1%) to avoid activation, the 10% FBS in the medium used in the present study could not affect the basal activation status of this cell line?

Precision-cut liver slices section:  could the authors mention how many different donors have been used for the preparations of the PCLS used for these analyses? It would be useful to report in this section which concentrations of TGF-b and PDGF-BB have been used for the stimulation of the PCLS.

Results

In the section reporting the modulation of fibrotic response authors refer to the differences in the gene expression in % vs unstimulated controls, while in the graphs (figure 2) the data is plotted as folds of induction. In order to help the interpretation would be useful to unify the criteria.

Regarding the expression of some genes, for instance, CCL5 and PPARG in primary HSC; or CCL5 in TWNT-4: from table S1 I can infer that the expression levels were undetectable and that TGF-b and PDGF-BB stimulation did not induce any difference. If this assumption is correct would be useful to mention it in the main text.  

Figure 2C and 2H: Authors are requested to put the respective cell names on the y-axis of the graphs.

Figure 4C: PCLS should be added in the y-axis

Supplementary Figure S1 C and G: please add the cell line’s name in each y-axis

Round 2

Reviewer 1 Report

the authors addressed all points in their point to point response